

# Influence of mental energy on volleyball competition performance: a field test

Shiow-Fang Shieh[1], Frank J.H. Lu[2], Diane L. Gill[3], Chih-Hsuan Yu[2], Shu-Ping Tseng[4] and Meisam Savardelavar[5]

[1] Graduate School of Leisure and Exercise, National Yunlin University of Science and Technology, Yunlin, Taiwan
[2] Graduate Institute of Sport Coaching Science, Chinese Culture University, Taipei, Taiwan
[3] Kinesiology, University of North Carolina at Greensboro, Greensboro, NC, United States of America
[4] Tainan University of Technology, Tainan University of Technology, Tainan, Taiwan
[5] Department of Sport Sciences, Zand Institute of Higher Education, Shiraz, Iran

## ABSTRACT

Athletic mental energy is a newly emerging research topic in sport science. However, whether it can predict objective performance in competition remains unexplored. Thus, the purpose of this study was to examine the predictability of mental energy on volleyball competition performance. We recruited 81 male volleyball players ($M$ age = 21.11 years $\pm$ SD = 1.81) who participated in the last 16 remaining teams in a college volleyball tournament. We assessed participants' mental energy the night before the competition and collected their competition performance over the next 3 days. We used six indices of the Volleyball Information System (VIS) developed by the International Volleyball Federation (FIVB) to examine its associations with mental energy. All six factors of mental energy –motivation, tirelessness, calm, vigor, confidence, and concentration correlated with volleyball competition performance. Further, a hierarchical regression found mental energy predicted volleyball receivers' performance ($R^2$ = .23). The findings advance our knowledge of mental energy and objective performance in competition. We suggest that future studies may examine the effects of mental energy on different sports with different performance indices.

Corresponding author
Frank J.H. Lu, frankjlu@gmail.com

## INTRODUCTION

Energy takes many forms such as potential energy, kinetic energy, thermal energy, electrical energy, chemical energy, nuclear energy, and other forms (*The New Encyclopedia Britannica, 1973*). We use energy to produce all sorts of products to satisfy our needs (*Wrigglesworth, 1997*). In the same manner, we eat different foods such as fruits, vegetables, grains, and meats to transfer into energy so we can move, work, and engage in all activities. Athletes transfer carbohydrates, fats, and proteins derived from foods into high-energy compounds such as adenosine triphosphate (ATP) and creatine phosphate (CP) so they can engage in all vigorous training and competition (*McArdle, Katch & Katch, 2006*).

In addition to physical energy, sports scientists are also interested in mental energy. Mental energy is defined as "…*an individual's ability to continue long hours of thinking,*

*concentrating attention, and blocking distractions to achieve a given task*'' (*Lykken, 2005*, p.331). As human beings, it is important to have sufficient mental energy to maintain attention and accomplish daily activities and work. Sports science researchers are also interested in how mental energy influences athletic performance. For example, in his book entitled ''leadership: full engagement for success,'' *Loehr (2005)* proposed a pyramid model of athletic success. *Loehr (2005)* contended that to achieve peak performance, athletes need four types of mental energy. At the bottom level, physical energy provides the basic needs of human function. The second level, ''emotional energy,'' guides physical energy. Then, the third level is mental energy which regulates physical energy and emotional energy. The top level, spiritual energy, guides all types of energy. *Loehr (2005)* suggested that mental energy is in a central role in all energies which govern athletes' higher level of functioning such as perception, abstract reasoning, creativity, memory, attention, motivation, self-awareness, and self-regulation. Without sufficient mental energy, athletes are unable to achieve their best performance (*Noakes, 2000*).

*Suinn (1986)*, the former president of the American Psychological Association (APA), developed psychological skill training for athletes called ''visual motor behavioral rehearsal (VMBR).'' *Suinn (1986)* proposed that every athlete has the potential to achieve their best if they can guide his/her energy by VMBR. After mastering the skills of VMBR, *Suinn (1986)* contended that athletes can regulate energies in competition. By doing so, athletes will come into a state of calm, confidence, and concentration so they can achieve their best. Despite these early researchers' interest in athletes' mental energy, what is mental energy, and whether mental energy really influences athletes' performance remains unknown.

Research in mental energy starts from nutrition studies in which researchers examined whether consuming sucromalt improves adults' mental and physical energy (*e.g.*, *Dammann et al., 2013*); or whether intake tryptophan-rich protein hydrolysate improves emotional processing, mental energy, and reaction (*e.g.*, *Mohajeri et al., 2015*); or consume Ginkgo biloba improves memory and mental energy (*e.g.*, *Snitz, O'Meara & Carlson, 2009*). Nutrition scientists adopted diverse definitions of mental energy such as attention ability (*e.g.*, *Mohajeri et al., 2015*; *Snitz, O'Meara & Carlson, 2009*), reaction time (*Mohajeri et al., 2015*), memory (*e.g.*, *Kennedy et al., 2007*), executive function (*Snitz, O'Meara & Carlson, 2009*), or emotional experiences (*e.g.*, *Mohajeri et al., 2015*). Further, researchers adopted all sorts of measuring tools such as memory tests (*Kennedy et al., 2004*), attention tests (*Kennedy et al., 2004*), mood tests (*e.g.*, *Johnson et al., 2008*), visual analogue scales (*Kennedy et al., 2004*), or self-developed questionnaires (*e.g.*, *Dammann et al., 2013*) to evaluate mental energy. Because of such diverse definitions and measures, *Lu et al. (2018)* adopted the framework of mental energy proposed by the North American Branch of the International Life Science Institute (ILSI) to define athletic mental energy as ''*…an athlete's perceived existing state of energy which is characterized by its intensity in motivation, confidence, concentration, and mood.*'' *Lu et al. (2018)* used six studies to develop a sport-specific mental energy measure termed the Athletic Mental Energy Scale (AMES), which is comprised of 6-factors including motivation, concentration, confidence, vigor, tirelessness, and calm. In Study 1, they recruited 11 experts in sports and psychology to form a focus group to establish the initial framework of athletic mental energy (AME). Study 2 sampled

242 college student-athletes to investigate their experiences of mental energy and develop a scale draft titled "Athletic Mental Energy Scale (AMES)." In Study 3, they recruited 243 college student-athletes to examine the psychometric properties and the underlying structure of AMES *via* item analysis, internal consistency, and exploratory factor analysis (EFA). In Study 4, they sampled another 312 college student-athletes and used confirmatory factor analysis (CFA) to examine AMES's factorial validity, concurrent and discriminant validity. In study 5, they examined the measurement invariance of the 6-factor, 18-item AMES with 223 Taiwanese and 156 Malaysian samples. In Study 6, they sampled 78 Malaysian martial artists to examine the predictive validity of AMES by comparing AMES scores and successful and unsuccessful competition outcomes. *Lu et al. (2018)* reported that AMES has sufficient content validity, factorial structure, nomological validity, discriminant validity, and predictive validity.

There are several distinctions between AMES and traditional sport psychology measures. First, it represents sports performers' "perceived energy" state which is manifested by emotional (*i.e.,* vigor, tirelessness, and calm) and cognitive components (*i.e.,* motivation, confidence, and concentration). Traditionally, sport psychology researchers use diverse emotional measures such as the profile of mood state (POMS, *McNair (1992)*) or competitive state anxiety inventory-2 (CSAI-2, *Lane et al., 1999*) to examine their relationships with performance. The AMES provides a simple method to assess athletes' emotional states by one scale. Further, sport psychology researchers may use several psychological measures such as State Sport Confidence Scale (SSCI, *Vealey, 1986*), Mindful Awareness Attention Scale (MAAS, *Brown & Ryan, 2003*), or Sport Motivation Scale-II (*Pelletier al., 2013*) to measure sports performers' confidence, attention, and motivation. Again, the AMES achieves these goals by a single measure. Thus, the AMES is not only being used to assess athletes' existing state of mental energy but also to measure sports performers' emotional and cognitive states at one time.

After the AMES was developed, researchers used it to examine its influence on athletes' behavior and performance. For example, to examine the relationship between nutrition intake and athletes' mental energy, *Yildiz et al. (2020)* sampled 254 Turkish athletes and administered them with AMES and the Three-Factor Eating Questionnaire (TFEQ-R18, *Karlsson et al., 2000*). They found positive eating behavior (*e.g.,* cognitive restraint) positively correlated with mental energy while negative eating behavior (*e.g.,* emotional eating, uncontrollable eating) was negatively correlated with athletes' mental energy.

In another study, *Chiou et al. (2020)* hypothesized that athletes' mental energy is a positive asset that might influence athletes' life stress-burnout relationship. They used two different samples (one is soccer, the other is multi-sports) to examine the triangular relationships among athletes' life stress, burnout, and mental energy. Results showed athletes' mental energy negatively correlated with life stress and burnout. Also, athletes' mental energy moderated athletes' life stress-burnout relationship in both samples.

Despite these initial findings, whether athletes' mental energy influences competition performance needs further examination. Recently, *Chuang et al. (2022)* conducted two studies to examine the associations between athletes' mental energy and competition performance with physically disabled tennis players. In the first study, they adopted an
ideographic approach to examine the relationship between the mental energy of nine elite physically disabled tennis players and their competition performance. They found each elite physically-disabled tennis player showed different levels of mental energy, and the associations between pre-competition mental energy and performance were personal and individualized. Study 2 sampled 77 national-level physically-disabled table tennis players and examined the pre-competition mental energy-performance relationship. Results showed athletes' performance correlated with all factors of mental energy- vigor, calm, tirelessness, motivation, confidence, and concentration. Further, a hierarchical regression analysis found confidence in mental energy predicted participants' "subjective" performance where participants rate their performance by responding to a question: "how do you feel about your performance in the last game?" with a score ranging from 0 to 100 points. (*Chuang et al., 2022*; p. 6).

Sport psychology researchers used a self-referenced measure (*e.g., Beedie, Terry & Lane, 2000*; *Lane & Chappell, 2001*; *Prapavessis, 2000*) to examine the effect of mood on athletes' performance. The measure asked participants to rate their competition performance by asking one question "How do you feel about your performance in the last game?" with a score ranging from 0 to 100 points. This measure is questionable because it can't reflect the true performance in the competition. To fill the gap, we examined the influences of athletes' mental energy on competition performance using an "objective" measure. Specifically, we recruited college volleyball players who competed at the national level in the semi-quarter final. To avoid disturbance, we measure their mental energy one day before the tournaments. Further, we used the Volleyball Information System (VIS) developed by the International Volleyball Federation (FIVA) to assess volleyball players' performance. The VIS provides six objective performance indices during the competition including top spiker, top receiver, top blocker, top digger, top setter, and top server. The VIS has been proven with appropriate content validity (*Yudiana et al., 2017*) and has been used in many empirical studies (*Marcelino et al, 2009*; *Valladares, García-Tormo & João, 2016*). For example, in a study that examined the main factors affecting female volleyball players' performance in the 2014 world Championship, *Valladares, García-Tormo & João (2016)* analyzed 102 games of the 24 participating teams through VIS. Results found victories were serving hits, excellent reception, serve reception, opponent serve errors, and serve faults. Similarly, to examine whether home *vs* guest teams win more competition, *Marcelino et al (2009)* analyzed 275 sets in the 2005 Men's Senior World League and 65,949 actions through VIS. Results showed that winning a set is significantly related to performance indicators (*i.e.,* set, reception, spike, serve, dig, and block) and home teams always have more probability of winning the game than away teams. Thus, it seems that VIS is an objective tool to collect volleyball players' performance. The purpose of this study was two-folded. First, we examined the relationship between the self-assessment score of mental energy and six objective defensive and offensive volleyball performance indices of volleyball performance without considering teammates' and opponents' influences. Second, we examined whether mental energy predicted volleyball competition performance. We hypothesized that the self-assessment score of mental energy will correlate with six objective volleyball performances, and the global mental energy will predict volleyball performance.

## METHODS

### Participants

Originally, 131 college volleyball players participated in a national tournament at the semi-quarterfinal (16 teams). Their competition positions include spikers, setters, blockers, diggers, and free players. At the first contact, two teams refused to participate in the study, thus only 14 teams with 89 players remained. After data screening, eight players' questionnaire packages were incomplete so only 81 participants were included in this study. Their mean age was 21.11 years (SD = ±1.81), with 8.65 years of sport participation (SD = ± 1.81), and 2.63 h of daily training (SD = ± 0.71).

### *Procedures*

Before data collection, we gained research ethical permission from Antai- Tian-Sheng Memorial Hospital Institutional Review Board (TSMHIRB-2-R-030−2.1). After approval, the first author contacted each team's manager and told them about the research project. If the team agreed to participate we met players in the meeting room near the competition arena. We informed them about the purposes of the study, the procedures, and their rights as participants. After the briefing, participants who were interested signed the consent form and completed a survey package with a demographic questionnaire and mental energy scale. It took about 10 min to complete the questionnaire package.

During the tournament, which was held for 3 days, we set up two JVC JY-HM90 cameras which have full HD 1,920 × 1,080 50p/50i to record players' performance on each side of the court. Two assistants monitored the recording process. One camera was used as a formal recording machine while the other one was for backup. Because the tournament adopted a single-elimination system, we recorded eight competitions on the first day, four competitions on the second day, and three competitions on the last day. A total of 15 recorded films were analyzed. Because analyses of volleyball performance need experienced experts to do it, so we invited eight retired volleyball players to analyze the films. Eight analyzers had at least four years of experience in college volleyball training and competition. They understand volleyball technical terms and were familiar with each specific volleyball skill. So we invited them to help us to analyze the volleyball competition performance. Before the formal analyses, we held a workshop on how to identify the objective performance provided by FIVA using a 30-minute film rehearsal. To make sure they analyzed the film correctly, they need to repeat several times if they have different opinions. After reaching 100% consistency, two reviewers worked together to analyze the 15 recorded films. First, they identified the jersey number of the player. Then, they identified the objective performance of the VIS. Each valid performance was confirmed by the other. If two reviewers had doubts about certain data, they replayed the film and made the final decision. Generally, a 2-hour match took 6 to 7 h to analyze. The measures and the identification of the VIS are described in the following section.

### *Measures*

*Demographic questionnaire.* The demographic questionnaire collected participants' ages, competition positions, jersey numbers, daily training hours, frequency of training per week,

and years of athletic experience. The participants' ages, daily training hours, frequency of training per week, and years of athletic experience were used to control confounding factors while doing the hierarchical regression. The demographic questionnaire about competition positions and jersey numbers were to identify and double-check their competition performance during data analyses.

*Athletic Mental Energy Scale (AMES).* We used the *Lu et al. (2018)* 18-item AMES to assess participants' mental energy. The 18-item AMES is comprised of six factors with three items in each factor as follows: (a) confidence (sample question as: "I feel I can win all competitions in the future"), (b) concentration (sample question as: " There's nothing distracting me in training and competition"), (c) motivation (sample question as: "I feel excited in future competitions"), (d) tirelessness (sample question as: "I feel there is an endless energy coming from my body"), (e) vigor (sample question as: "I feel spiritual to do everything in sports"), and (f) calm (sample question as: "When facing to my opponents I am calm"). To calculate each item, AMES uses a 6-point Likert scale that ranged from 1 (not at all) to 6 (completely so) to evaluate participants' responses about their mental energies.

*Measures of performance.* We used the Volleyball Information System (VIS) provided by FIVB to assess participants' objective performance. The VIS has six indices to assess player's performance during competition as follows:

(a) Top spiker: This is calculated by summing the number of points by spike minus the number of faults, then dividing by the total number of attempts.
(b) Top blocker: The top blocker is defined as the player with the most kill blocks on average per set played by the team. To calculate this index, we summed the number of points by block and minus the number of faults by it, then divided by the total number of attempts.
(c) Top server: The top server is defined as the player with the most aces on average per set played by the team. To calculate this index, we summed the number of points by serve minus the number of faults, then divided by the total number of attempts.
(d) Top digger: The top digger is defined as the player with the most digs on average per set played by the team. The dig is defined as an action to defend the spike of the opponent. To calculate this index, we summed the number of successful digs minus the number of digging mistakes, then divided by the total number of attempts.
(e) Top setter: The top setter is the player with the most running sets on average per set played by the team. To calculate this index, we summed the number of successful sets minus the number of mistakes that resulted in opponent scores, then divided by the total number of attempts.
(f) Top receiver: The top receiver is defined as the number of successful receptions minus the number of faults divided by the total number of attempts.

## Statistical analyses

Before data analyses, we checked the raw data including completed questionnaires and recorded performance. Those players who did not complete the questionnaires or have

incomplete competition records (*i.e.,* injuries or withdrawal) were deleted. Further, we screened all data by examining means, standard deviations, skewness, kurtosis, and outliers to make sure there were no abnormal data. Thus, only 81 out of 89 players' data were qualified for statistical analyses. We used SPSS for Windows 22.0 software to analyze data. In this study, the independent variables were mental energy and its sub-factors; the dependent variables were the 6 objective indices of VIS. We used descriptive statistics to determine the variables' mean, standard deviation, frequency, and Cronbach $\alpha$ coefficients of the AMES. We used Pearson correlation analysis to examine the bivariate relationships among factors of mental energy and volleyball performance. Furthermore, we used hierarchical regression to examine the predictability of mental energy on objective indices of VIS. To perform this analysis, we entered the volleyball index of the VIS as a criterion variable (*e.g.*, top spiker). Then, we entered predictors. In the first model, we entered the aforementioned age, years of experience, and training hours to control potential confounding factors. Then, we summed all factors of mental energy as a composite score (*i.e.,* volleyball players' mental energy), and entered it in the second model. Thus, there are six separate hierarchical regressions were performed (*i.e.,* top spiker, top setter, top digger, top receiver, top sever, and top blocker).

## RESULTS

### Descriptive statistics

The highest mental energy scores were concentration, confidence, and vigor. The lowest scores were calm, motivation, and tirelessness. For volleyball performance, the highest scores were top setter, top scorer, and top digger. In contrast, the lowest scores were top server, top blocker, and top receiver. For internal consistency, mental energy factors ranged from .69 to .85, all in an acceptable range.

### *Bivariate correlation*

For the relationship between mental energy and volleyball performance, confidence, motivation, concentration, and vigor of mental energy correlated with the top receiver; tireless correlated with the top spiker, and calm correlated with the top digger. For the intra-relationships among mental energy, confidence correlated with motivation, concentration, tirelessness, calm, and vigor; motivation correlated with concentration and vigor; concentration correlated with tirelessness and vigor; tirelessness correlated with vigor. For intra-relationships among volleyball performance, the top receiver correlated with the top digger, and; the top blocker correlated with the top server (see Table 1).

### *Prediction of mental energy on objective volleyball performance*

We used six indices of VIS as criterion variables in the hierarchical regression and examined the predictability of mental energy on these indices. Because age and training hours are found to influence athletic performance (*Armstrong, Barker & McManus, 2015*; *Haff & Nimphius, 2012*), so we entered age and training hours in the first step. Further, because mental energy is characterized by having strong vigor, calm, tirelessness, motivation, confidence, and concentration, so adding all mental energy factors makes a composite
**Table 1  Descriptive statistics and bivariate matrix of the study variables.** This table illustrates basic statistical information of the study variables.

|  | Confi | Motiv | Concet | Tire | Calm | Vigor | Receiv | Spike | Block | Serve | Set | Dig |
|---|---|---|---|---|---|---|---|---|---|---|---|---|
| confi | 1.00** |  |  |  | . |  |  |  |  |  |  |  |
| motiv | .26* | 1.00 |  |  |  |  |  |  |  |  |  |  |
| concent | .35** | .37** | 1.00 |  |  |  |  |  |  |  |  |  |
| tire | 32** | .16 | .28* | 1.00 |  |  | . |  |  |  |  |  |
| calm | .41** | .18 | .19 | .16 | 1.00 |  |  |  |  |  |  |  |
| vigor | .25* | .33** | .46** | .25** | .03 | 1.00 |  |  |  |  |  |  |
| receiv | .31* | .43** | .37** | .21 | .21 | .28** | 1.00 |  |  |  |  |  |
| spike | .01 | .16 | .18 | .27** | .09 | .19 | .16 | 1.00 |  |  |  |  |
| block | −.12 | −.04 | −.16 | −.10 | −.06 | −.06 | −.01 | .02 | 1.00 |  |  |  |
| serve | −.01 | .11 | −.13 | −.14 | .11 | −.06 | −.02 | .06 | .31** | 1.00 |  |  |
| set | −.06 | −.06 | −.01 | −.04 | −.02 | .10 | −.04 | −.09 | .00 | −.04 | 1.00 |  |
| dig | .01 | .17 | −.01 | −.03 | .21* | .01 | .22* | −.08 | −.02 | .10 | −.05 | 1.00 |
| M | 12.46 | 9.63 | 13.37 | 9.48 | 7.69 | 12.38 | .34 | .36 | .23 | .17 | 1.55 | .81 |
| SD | 2.46 | 1.95 | 3.19 | 3.22 | 2.53 | 2.75 | .34 | .21 | .41 | .30 | 4.03 | .87 |
| $\alpha$ | .71 | .69 | .80 | .85 | .84 | .75 | – | – | – | – | – | – |
| N | 81 | 81 | 81 | 81 | 81 | 81 | 172 | 211 | 264 | 264 | 264 | 264 |

Notes.
*$p < .05$.
**$p < .01$.

confi, confidence; motiv, motivation; concet, concentration; tire, tirelessness; receiv, top receiver; spike, top spiker; block, top blocker; serve, top server; set, top setter; dig, top digger; score, top scorer.

**Table 2  Prediction of mental energy on volleyball performance.** This table shows how mental energy predicts volleyball performance.

| Variable | Model 1 | | | Model 2 | | |
|---|---|---|---|---|---|---|
|  | B | SEB | $\beta$ | B | SEB | $\beta$ |
| Constant | 1.05 | .65 |  | −.28 | .70 |  |
| age | −.04 | .03 | −.19 | −.02 | .03 | −.13 |
| training hrs | −.00 | .02 | −.01 | −.01 | .02 | −.01 |
| Mental energy |  |  |  | .02 | .01 | .45 |
| $R^2$ |  | 0.04 |  |  | 0.23 |  |
| $F$ for change in $R^2$ |  | .85 |  |  | 11.45** |  |

Notes.
**$p < .01$.

predictor, mental energy.
criterion: top receiver.

score as mental energy. After six separate analyses, it was found that mental energy predicted top receiver. As Table 2 shows, the first model found age, years of experience, and training hours had no effects on prediction. In the second model, mental energy significantly predicted the top receiver ($R^2 = .23$).

## DISCUSSION

The purpose of this study was to examine whether athletes' mental energy influences objective performance in sports. To achieve this objective, we recruited 81 volleyball players

and examined the associations between mental energy and objective volleyball performance in competition. Results found all factors of mental energy-motivation, tirelessness, calm, vigor, confidence, and concentration correlated with volleyball competition performance. Further, a hierarchical regression found mental energy predicts the top receiver. Results filled a gap in knowledge of athletes' mental energy-performance relationships. The initial results provided several implications for researchers.

## Theoretical contributions/implications

From *Lykken (2005)* to the North American Branch of the International Life Science Institute, all researchers consider mental energy important for human work and performance. We adopted Lu et al.'s (*2018*) concept and measure to examine whether mental energy can influence volleyball competition performance. Because *Lu et al. (2018)* contended that athletic mental energy is comprised of emotional and cognitive components. We discuss our results following those two parts.

### *Influence of emotional components of mental energy on volleyball performance*

For a long time, researchers have been interested in how athletes' emotions influence sport performance (*Janelle, Fawver & Beatty, 2020*). Emotions are the brief affective states elicited in response to environmental stimuli, which in turn, trigger psychophysiological responses to react (*Lewis, Haviland-Jones & Barrett , 2010*). Athletes' emotions lead to changes in physiological responses (*e.g.*, muscle tones, heart rate, breath), attention, and cognitive function (*e.g.*, information processing), and consequently influence athletic performance (*Gill, Williams & Reifsteck, 2017*). Our study found vigor correlated with the top receiver, calm with the top digger, and tirelessness associated with the top spiker are theoretically meaningful. First, vigor is defined as "…an individual's subjective feeling with heightened arousal (*Lane & Terry, 2000*). It is important for volleyball players to maintain high vigor so they can perform well during competition. Our study not only supports a recent meta-analysis that found the effect size for the vigor-athletic performance is low to moderate (*Beedie, Terry & Lane, 2000*; *Lochbaum et al., 2021*) but also support Chuang et al.'s study (*2022*) that found athletes' emotional components of mental energy (*i.e.,* vigor and tirelessness) associated with table tennis performance. The vigor-volleyball performance relationship may be related to the tirelessness-volleyball performance relationship because vigor and tirelessness are the same emotional factor. It is because Lu et al.'s (*2018*, p. 6) exploratory factor analysis (EFA) separated all items of tirelessness from the factor of "vigor."

Further, our finding that calm correlated with a top digger is insightful both in volleyball and sports psychology research. In volleyball research, *Jimenez-Olmedo & Penichet-Tomás (2017)* contended that diggers not only have to work with blockers to defend field zones but also build a counterattack, and analyze the position and placement of the player who makes the attack (*Hernández et al., 2004*). Thus, diggers must possess calm attributes so they can maintain a clear mind to decide what to do and where the ball is to go. On the other hand, sport psychology researchers contend that when athletes are in a state of peak performance, they were characterized as having "no fear of failure" and are "physically and mentally

relaxed" (*Lohr, 1984*, p. 67), which are similar to clam. Further, *Burns, Weissensteiner & Cohen (2019)* interviewed 10 Olympic, Para-Olympic, and world champions about their mindset and lifestyle and found participants reported that staying calm during training and competition is one of the key factors of their success. Thus, our study echoed these studies in elite sport psychology/volleyball research and supports that calm plays an important role for volleyball players during competition. Despite these initial findings, we still found emotional components of AMES didn't correlate with some indices of VIS (*i.e.,* block, set, and dig). The reasons are unknown. We will discuss it in limitations.

### Influence of cognitive components of mental energy on volleyball performance

We found three cognitive components of mental energy (*i.e.,* concentration, confidence, and motivation) correlated with top receivers in volleyball competitions. Much research contends that to perform well during competition, athletes must maintain high concentration so they won't be distracted by irrelevant stimuli (*Williams et al., 2015*). Concentration is the ability to pay attention to what is most important in any situation while ignoring distractions (*Moran, 2008*). Further, some elite athletes reported that they were "…*able to focus on tasks at hand,*" or "*feeling completely detached from the external environment and any potential distractions*" when they achieved peak performance (*Garfield & Bennett, 1984*; *Williams et al., 2015*). Thus, the significant concentration-top receiver relationship in our study is consistent with previous research.

Similar to the emotional components of AMES-VIS relationship, we did not find any correlation between other cognitive components such as motivation and confidence with volleyball competition performance. In *Chuang et al. (2022)* found all cognitive components of mental energy (*i.e.,* concentration, motivation, and confidence) were correlated with physically disabled table tennis players' subjective performance. The reasons are unknown. It is plausible that using objective VIS performance indices may have shadowed some significant mental energy-volleyball performance relationships. We discuss this limitation in the next section.

## Limitations and future suggestions

There are several limitations should be noted. First, because of the huge time demand for video analyses, it was difficult to recruit a large sample. We suggest future studies combine similar data sets to increase statistical power (*Akobeng, 2016*). Second, to avoid distraction before the competition, the participants' coaches only agreed to measure athletic mental energy one night before the tournament. We did not measure participants' mental energy immediately before the competition. Whether our measurements reflect participants' existing state of mental energy in competition needs further examination. Third, we only recruited volleyball players as our participants; whether our results apply to other sports such as weight-lifting, golf or archery needs further examination. Fourth, we recruited college student-athletes as participants, whether our results apply to higher-level athletes (*e.g.,* professional) or younger participants needs to be further examined. Fifth, we didn't monitor participants' training sessions before and during the competition, whether their training conditions influenced competition performance needs further examined.

Moreover, we used the official VIS of the FIV to evaluate participants' competition performance. The VIS has some limits to linking participants' performance and mental energy. First, if the performer's errors are caused by coaches' instructional mistakes, the VIS equation still includes it as the performer's failure. Further, some performers' successes are caused by opponents' mistakes but are still counted as participants' successes. Furthermore, the total number of attempts of each volleyball performance (*e.g.*, spike or receive) during competition processes mixed with teammates' mistakes/successes. The blend of players' performance with others makes the denominator in the VIS's equation (total attempts of each performance) very large, which attenuates the volleyball performance index in VIS. In addition, although we have carefully used a group of experienced experts to analyze all recorded videos of the competition, we didn't use alternate software such as *DataVolleyball (2023)* to analyze our data. We suggest future studies may use this software to do similar studies.

### Strengths of the study

Many studies attempted to examine the effects of psychological factors on athletic performance using subjective performance as the criterion (*e.g.*, *Beedie, Terry & Lane, 2000*; *Lane & Chappell, 2001*; *Prapavessis, 2000*). This provides vague information about the association between psychological factors and real performance. To fill this gap, we used seven indices of the Volleyball Information System developed by FIVA and examined its associations with mental energy. Results found both emotional and cognitive components of mental energy associated with volleyball competition performance. *Martens (1987)* contended that to expand our knowledge in sports science we need to move our research from the laboratory to the field. Field settings have more variants than laboratory experiments but they reflect the real world of sports; especially the interactions of the audience, opponents, referees, officials, and teammates. Through field research, we examined whether mental energy is associated with volleyball competition performance. Our results not only advance our knowledge about how mental energy influences volleyball competition performance, but also indicate which factors of mental energy contribute to volleyball competition performance. We encourage more researchers to adopt this approach to examine the mental energy-performance relationship.

## CONCLUSION

Results found all mental energy components–motivation, tirelessness, calm, vigor, confidence, and concentration correlated with volleyball competition performance. Also, global mental energy predicts volleyball receivers' performance. Our results support hypotheses and provide initial knowledge on this topic. We encourage more researchers to examine the mental energy-performance relationship in different sports and contexts in the future.

## Funding

This project is funded by the Ministry of Science and Technology, MOST, Taiwan (MOST 108-2410-H-034-042). The funders had no role in study design, data collection and analysis, decision to publish, or preparation of the manuscript.

## Grant Disclosures

The following grant information was disclosed by the authors:
Ministry of Science and Technology, MOST, Taiwan:  MOST 108-2410-H-034-042.

## Competing Interests

Frank JH Lu is an Academic Editor for PeerJ.

## Author Contributions

- Shiow-Fang Shieh conceived and designed the experiments, analyzed the data, authored or reviewed drafts of the article, and approved the final draft.
- Frank J.H. Lu conceived and designed the experiments, authored or reviewed drafts of the article, and approved the final draft.
- Diane L. Gill conceived and designed the experiments, prepared figures and/or tables, and approved the final draft.
- Chih-Hsuan Yu performed the experiments, authored or reviewed drafts of the article, and approved the final draft.
- Shu-Ping Tseng performed the experiments, analyzed the data, prepared figures and/or tables, prepare video cameras, and approved the final draft.
- Meisam Savardelavar performed the experiments, analyzed the data, prepared figures and/or tables, and approved the final draft.

## Ethics

The following information was supplied relating to ethical approvals (i.e., approving body and any reference numbers):

Antai-Tian-Sheng Memorial Hospital Institutional Review Board

## Data Availability

The raw data is available in the Supplementary File.

## Supplemental Information

Supplemental information for this article can be found online at http://dx.doi.org/10.7717/peerj.15109#supplemental-information.

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
