# Peer review of "Influence of mental energy on volleyball competition performance: a field test"

_PeerJ, doi:10.7717/peerj.15109_

## Round 0.1 · original submission · Major Revisions

Dear Authors

The manuscript has been reviewed by three expects in the field of the study. We believe your study provides valuable input for our current understanding of applied sports psychology and volleyball performance. The reviewers have addressed several issues for the next round of revision. We would like to invite you to submit a revised version of the manuscript that addresses the points raised by the reviewers, particularly regarding the definition of mental energy and its association with volleyball performance during competitions.

We look forward to receiving your revised manuscript.

Best regards

Yung-Sheng Chen, PhD
Academic Editor

Reviewer 1 ·

Basic reporting

This study examined whether mental energy can be considered a potential predicative factor of real-world sporting performance in a sample of college volleyball players. The authors recruited 81 volleyball players and assessed their mental energy one night before the competition and collected their sporting performance during the games. The main findings are that all sub-components of mental energy correlated with some specific sporting performance. More specifically, the hierarchical regression revealed the predictive power of mental energy in determining the receivers’ performance. Based on these findings, the authors suggested the utility of mental energy in real-world sports settings.

In my opinion, the manuscript is interesting and well-written, and the topic is of interest for PeerJ. However, there are some major caveats of this study which are needed to be addressed appropriately.

1. More details of the Athletic Mental Energy Scale (AMES) by Liu et al (2018) are needed, especially for the emotional and cognitive components of mental energy that provide basis for the discussion of the present findings. Otherwise, it is difficult to understand what may be the difference between AMES and other traditional psychological measures.

2. Line 92, what did you mean by “subjective performance” and how to measure in this study? Please specify.

3. It is suggested to provide appropriate citations to justify the validity of Volleyball Information System (VIS). Doing so may strengthen authors’ argument regarding the importance of “objective measure”.

4. Please make specific research hypothesizes in line with the research purposes.

5. Line 140 – 142, it could be just me missing the information in the manuscript, but it was not clear to me why only some of the demographic information were considered as confounding factors for the subsequent analysis? For example, it can be assumed that competition position may also significantly affect the performance.

6. For replications purposes, it would be interesting to point out how to perform the procedure of deduction of invalid questionnaires.

7. Please provide more information (i.e., the rationale for the analytic strategy) about the hierarchical regression. Further, why and how seven separate analyses were performed? All the results of these analyses should be reported.

8. In the discussion, although the authors have provided possible interpretations regarding the findings, some null effects and the inter-relationships across variables among the AMES and VIS are worthy to mention.

Experimental design

Please see the general comments.

Validity of the findings

Please see the general comments.

Additional comments

Please see the general comments.

·

Basic reporting

PeerJ review
Title: Influence of mental energy on volleyball competition performance: A field test

Dear Authors, I appreciate Your work on this interesting topic. In this review I tend to help You increase the quality of the work.

Line 30 – so called by the Authors "hydraulic energy" is basically the potential and kinetic energy of moving water, not some separate type of energy. Forms of energy:
mechanical energy
-kinetic energy
-potential energy
-elastic energy
Energy of the electromagnetic field:
-electricity
-magnetic energy
chemical energy
nuclear energy
energy of thermodynamic potentials
I recommend either searching for reliable reference from physics or cutting out this sentence and reframing the introduction.
No matter which kind of food or macronutrients we eat they contain the same kind of energy – chemical energy.
What is more organism do not directly “transfer” macronutrients to ATP or PCr but mainly use the energy stored in food to produce ATP and PCr from the ingredients it already have.
Continuing this thought, “mental energy” can not be treated as another form of energy but rather as activity, intensity or some other abilities measured by score in tests used to assess analyzed by the authors features: motivation, concentration, confidence, vigor,tirelessness, and calm.
I understand that authors' own research is quoted, but I strongly recommend to show to the readers different views coming from psychological scientific literature describing ideas of “mental energy” because different authors mention different features in the introduction and use different therms like personality traits, mental characteristics or psychical abilities or capacity.

English is professional throughout the manuscript, but in several places colloquial or expressing a personal opinion words are used – f.e. line 41 – inspiring,
Line 68 – for examine, to examine – shouldn’t it be for example – to examine ?
Line 146 - There’s nothing distracting me in training and training – probably in training and playing.
148 and 149 – the same sample sentences for different features
Lines 209-211 – repetition from the methods section
Line 290 – vey ?

I recommend double check the English for grammar and professional English by a colleague who is familiar with this topic.

Experimental design

I recommend changing title after reconsidering the aims, results and conclusions
Possible title:
Relationship of self-assessment mental capacity score AMES and performance in aspect of reliability measured by VIS in college volleyball players during a teournament.

The research questions/aims are not well defined.
Considering that this study analyses self-assessment score in relation to performance in aspect of reliability in defensive and offensive volleyball actions assessed in a way neglecting the impact of teammates and opponents influence the goals set by the authors were incorrectly formulated.
What is more to be honest it should be precised that the measured by self assessment mental energy level is one day before the competition which can be probably important because as the authors surely know the mental state of the players can change significantly just before the game and also differ from the morning before the game and a day before a game.
Line 50 an 58– This statements exclude each other, so can You give a reference for this words or is this a hyphothesis (line 50)?

Validity of the findings

Table 1:
Misspelling – concent – concet
In some places there are * put to the 1.00 for the same factor – f.e. tire

the provided files containing data are correct.

Conclusions are not well stated, they are not linked with research questions nor limited only to supporting results.
In my view the conclusions section should be all reframed. The conclusions section must include only answer to the questions of the study or a sentence confirming or declining the stated hyphothesis. Conclusions should be consistent with properly formulated aims of the study and should not contain any subjective sentences opinions etc.

Additional comments

The paper is interesting but need a lot of work to increase quality.

·

Basic reporting

no comment

Experimental design

Original primary research within Aims and Scope of the journal.
Research question well defined, relevant & meaningful. It is stated how research fills an identified knowledge gap.
The submission clearly define the research question, which is relevant and meaningful. The knowledge gap being investigated is identified, and statements were made as to how the study contributes to filling that gap.
Rigorous investigation performed to a high technical & ethical standard.
In the Methods section, (line 131) why did you opt to choose the retire players to analyse the videos instead the performance analist/ scouter of volleyball teams. In line 133 you wrote "100% consitency" - is it possible to get that value? What werethe main problems regarding the performance analysis?
The measure of performance (VIS) was validated and/or has reliability? Please add some reference with that validation. Why didnt you use the Data Volley (there are several studies using Data Volley to analyse the performance). Please define successful receptions (according to what? Setter position? Setter options do set the ball?). The variable Top Setter it's depending on the attacker efficacy (is that fair to analyse?).

Validity of the findings

no comment.

Additional comments

Dear Authors!
Congratulations for your valuable work.
Despite the well structure and novelty of your study, ther ar some recommendations that i would like to suggest in order to improve your work.
In line 11, please change FIVA to FIVB (as well as in all document).
In the Introduction section (line 59 to 66), please give more details abaout the study of Lu et al. (2018) (i.e. methology, participants...).
In line 220 and 221 please correct the reference of Lu et al. 20118 to 2018.
Another Limitation of your study could be the lack of monitoring of the training session according to the period analyzed (according to the literature, the conditioning of the athletes should predict the performance).

---

## Round 0.2 · Minor Revisions

Please address the additional reviewers comments

Reviewer 1 ·

Basic reporting

I appreciate the author's diligent revision and responses to the reviewers' feedback. It appears that the authors have carefully addressed all of the comments and concerns raised during the review process. Based on the quality of the revised manuscript, I would recommend its publication in PeerJ.

Experimental design

no further comments

Validity of the findings

no further comments

Additional comments

no further comments

·

Basic reporting

All my previous suggestions have been taken into account and corrected.

Experimental design

All my previous suggestions have been taken into account and corrected.

Validity of the findings

All my previous suggestions have been taken into account and corrected.

Additional comments

Conclusions section
I would kindly reccomend shortening and leaving only lines 464-468.
I would reccomend to put emphasis on the fact that the study was measuring mental state the day before the match - to do this put a clear sentence in the introduction section and give reasons that making it this way is in accordance with sports pratcice in which the players and coaches would like to concentrate only on match on the match day and do not feel comfortable doing additional measures during match day before the competition.

---

## Round 0.3 · accepted · Accept

Dear Authors, I would like to express my grateful thanks for your patience and efforts to improve the quality of the manuscript. Your submission is now endorsed by three experts for acceptance of publication in PeerJ. Congratulation!!!

Before the publication, please show 6 indices of the Volleyball Information System (VIS) in the abstract. Additionally, please demonstrate the range of statistical outputs for “All 6 factors of mental energy – motivation, tirelessness, calm, vigor, confidence, and concentration correlated with volleyball competition performance.” in the abstract as well.

Best Regards
Yung-Sheng Chen

·

Basic reporting

All my previous suggestions have been implemented and corrected.

Experimental design

All my previous suggestions have been implemented and corrected.

Validity of the findings

All my previous suggestions have been implemented and corrected.

Additional comments

I recommend further proceeding with this manuscript aiming for publication at PeerJ.

All the best to the Authors, Editor and reviewers.